# Expression Profiles of Microsatellites in Fruit Tissues of *Akebia trifoliata* and Development of Efficient EST-SSR Markers

**DOI:** 10.3390/genes13081451

**Published:** 2022-08-15

**Authors:** Wei Chen, Huai Yang, Shengfu Zhong, Jun Zhu, Qiuyi Zhang, Zhi Li, Tianheng Ren, Feiquan Tan, Jinliang Shen, Qing Li, Peigao Luo

**Affiliations:** 1State Key Laboratory of Plant Breeding and Genetics, Sichuan Agricultural University, Chengdu 611130, China; 2College of Forestry, Sichuan Agricultural University, Chengdu 611130, China; 3Department of Biology and Chemistry, Chongqing Industry and Trade Polytechnic, Chongqing 408000, China

**Keywords:** *Akebia trifoliata*, transcriptome, EST-SSR markers, expression profile, validation test

## Abstract

*Akebia trifoliat**a*, a member of the family Lardizabalaceae, has high exploitation potential for multiple economic purposes, so genetic improvements to meet requirements for commercial demand are needed. However, this progress is largely impeded by a lack of effective selection markers. In this study, we obtained 271.49 Gb of clean transcriptomic data from 12 samples (three tissues at four developmental stages) of *A. trifoliata* fruit. We identified 175,604, 194,370, and 207,906 SSRs from the de novo assembled 416,363, 463,756, and 491,680 unigene sequences obtained from the flesh, seed, and rind tissues, respectively. The profile and proportion of SSR motifs expressed in each fruit tissue and developmental stage were remarkably similar, but many trinucleotide repeats had differential expression levels among different tissues or at different developmental stages. In addition, we successfully designed 16,869 functional EST-SSR primers according to the annotated unigenes. Finally, 94 and 72 primer pairs out of 100 randomly selected primer pairs produced clear bands and polymorphic bands, respectively. These results were also used to elucidate the expression profiles of different tissues at various stages. Additionally, we provided a set of effective, polymorphic, and reliable EST-SSR markers sufficient for accelerating the discovery of metabolic and pathway-specific functional genes for genetic improvement and increased commercial productivity.

## 1. Introduction

*Akebia trifoliata* (Thunb.) Koidz, commonly called augmelon in China, is a domesticated representative of the Lardizabalaceae family used in crop cultivation. The growing area throughout East Asia, and especially in China, is rapidly increasing [1]. *A. trifoliata* can be exploited for multiple purposes, such as traditional Chinese medicine [2], delicious fruit [2,3], edible oil [4], and ornamental horticulture [5]. All commercial exploitation centers on the fruit, although other parts, such as stems and roots, have been used in various fields, such as cosmetics and other industries [5].

For example, the various beautiful shapes and attractive colors of whole *A. trifoliata* fruit possess aesthetic value, and *A. trifoliata* can be used as an ornamental plant [1]. However, the plant also produces many secondary metabolites, such as saponins and flavones, in the whole immature fruit and the rind of mature fruit, which have made it a traditional medicinal plant in East Asia, including China, Korea, and Japan, for at least 2000 years [2]. The flesh of mature fruit is delicious and contains many beneficial nutrients, such as sugars [6], free amino acids [3], vitamins [3], and minerals [7], which gradually make it a new fruit for consumption and health purposes [2,8]. Reports have also suggested that the mature fruit has many seeds that have a high oil content (usually above 45%), and the oil is rich [4] in saturated, monounsaturated, and polyunsaturated fatty acids, with a proportion close to 1:1:1 [9], which matches the composition of ideal oils recommended by an increasing number of authors [10] for nutritional and health purposes. Therefore, *A. trifoliata* also has enormous potential as a new edible oil crop. The economic value of various tissues of the *A. trifoliata* fruit has received much attention, and the rate of corresponding commercial demand and exploitation is increasing. Unfortunately, there has been little work on improving the genetics controlling fruit traits of commercial interest for increased production.

Various molecular markers, such as restriction fragment length polymorphisms (RFLPs), sequence-tagged sites (STSs), single-sequence repeats (SSRs), and single-nucleotide polymorphisms (SNPs), have been developed as powerful selection tools in many cereal crops to assist breeding [11,12]. In recent years, the application of molecular markers in horticultural crops has further promoted the identification of functional genes and population genetic studies in plants, such as *Paeonia* [13], *Rosa* [14], and melon [15], which has greatly accelerated the progress of genetic improvement and has largely enhanced the effectiveness of selection on new commercial cultivars. Among them, SSRs widely exist in intergenic and genic regions of both prokaryotic and eukaryotic organism genomes [16]. SSRs are usually abundant, codominant, reproducible, highly polymorphic, and genomewide, so they have commonly been used as powerful molecular tools in gene mapping, DNA fingerprinting, genotype or varietal identification, molecular marker-assisted breeding, and population diversity analysis [17,18,19]. The number of SSRs in intergenic regions is generally greater than that in genic regions in a given eukaryotic species, although the density of SSRs in some genic regions could be higher than the average SSR density in intergenic regions because genic regions compose only a small proportion of the entire genome compared with intergenic regions [20,21]. However, SSR markers in genic regions are more effective than those in intergenic regions for genetic improvement programs and the enhancement of economically interesting crop traits, especially in studies exploiting wild species used to improve cultivars because they were designed on the basis of gene-self sequences rather than on flanking gene sequences.

In addition, the original sequences used to detect SSRs can be divided into genomic SSRs and EST-SSRs (expressed sequence tag-SSRs) [22]. Because EST-SSR markers are derived from expressed sequences and located within genic regions, their polymorphism can be directly coupled with functional differences in the corresponding genes [23]. Practically, the EST-SSR markers derived from some genes involved in a specific biological pathway or metabolic process would play a larger role in molecular-assisted selection for a given objective trait influenced by that pathway than genome-wide markers [24]. For example, EST-SSR markers were found to be involved in disease resistance and pigment biosynthesis processes in *Triticum aestivum* and *Prunus salicina* [25,26]. In addition, the development of EST-SSR markers is relatively less time-consuming and less expensive than that of genomic SSR markers, especially for plant species without available reference genomes. Therefore, EST-SSR markers are more likely to be associated with specific phenotypic traits and are more valuable, more effective, and less expensive for molecular breeding and functional gene mining when exploiting wild plants for the genetic improvement of cultivars.

At present, Gene Ontology (GO) and Kyoto Encyclopedia of Genes and Genomes (KEGG) analysis are two especially useful and powerful methods to represent the detailed information of the genes or biological protein functions in well-known signal or metabolism pathways or processes in model species [27,28]. In addition, with the development and progress of DNA and RNA sequencing technology, the fast acquisition of transcriptome data from different experimental treatments and different developmental stages of various organs can be made available to scientists worldwide. It is evident that effectively focusing on a few useful genes and corresponding proteins involved in the signal and metabolism pathways related to important agronomic and economic traits via both GO and KEGG analysis is a powerful strategy to further develop expression sequence-based molecular markers, such as EST-SSR markers [29,30].

In this study, we acquired transcriptome data for three tissues, including the rind, flesh, and seed, of *A. trifoliata* fruit at four different developmental stages using Illumina HiSeq technology. These data were then employed to isolate and identify candidate unigenes containing SSRs. Both motif and expression profiles were outlined, and expression differences in trinucleotide repeats (TNRs) in different tissues at different stages were highlighted. Our objective was to develop and validate a set of highly effective EST-SSR markers, which could be valuable for improving traits of interest in *A. trifoliata* for specific commercial or economic purposes as well as for further exploitation of other wild plants.

## 2. Materials and Methods

### 2.1. Plant Materials and Sequencing

The clonal lines of “Shusen 1” were used to produce transcriptome data by Illumina sequencing because the available reference genome for *A. trifoliata* was also produced from this strain [31]. All clonal materials were cultivated in the *A. trifoliata* Germplasm Nursery of Sichuan Agricultural University at the Chongzhou Research Station (30°43′ N–103°65′ E) with a randomized complete block experimental design. Three healthy tissues, including the flesh, peel, and seed, of fruits at four distinct stages, including young, enlargement, coloring, and maturity, using the previously classified standard [1] were randomly collected from 2-year-old plants for transcriptome sequencing, and there were three biological replicates for each treatment. To construct sequencing libraries, total RNA from all 36 samples was extracted using TRIzol reagent (Invitrogen, Waltham, MA, USA) according to the manufacturer’s instructions. The paired-end libraries with 240 bp inserts were prepared and then sequenced on an Illumina HiSeq 2500.

In addition, a total of 45 *A. trifoliata* accessions were selected from germplasm nurseries according to geographic origin, distribution density, and phenotypic character to verify the polymorphism of EST-SSR markers; these strains included 15 selected super strains [3], 15 collected accessions from the Sichuan Basin as the putative origin of *A. trifoliata*, and 15 accessions from a large geographic scope of the natural distribution of *A. trifoliata*.

### 2.2. De Novo Assembly and SSR Identification

Raw transcriptome sequences were filtered by Trimmomatic (version 0.30, Jülich, Germany) software according to a previously described method [32]. Next, all high-quality reads were de novo assembled using Trinity (version 2.4.0, San Diego, CA, USA) tools [33], and the de novo assembled unigenes were further used to detect SSRs with the MicroSatellite (MISA) protocol [34]. To ensure the high quality of EST-SSR markers, the default parameters were set as follows: the minimum repeat numbers for single-nucleotide repeats (SNRs), dinucleotide repeats (DNRs), and other repeats, including trinucleotide repeats (TNRs), tetranucleotide repeats (TtNRs), pentanucleotide repeats (PNRs), and hexanucleotide repeats (HNRs), were 10, 6, and 5, respectively. In addition, the criterion for compound SSRs was defined if the interval between two SSR sequences was less than 100 bp.

### 2.3. Statistical Analysis of TNR Motifs

ANOVA (*p*-value = 0.05) was performed on the number of TNRs among the three tissues and four developmental stages of *A. trifoliata* fruit using SPSS 20 software. The chi-square test was also used to test whether the number of complementary TNRs of two strands fit a 1:1 ratio.

### 2.4. Functional Annotation of Unigenes

To identify the potential functions of unigenes carrying SSR loci, the sequences of unigenes were mapped to the GO and KEGG databases through the eggNOG-mapper (version 2.0.0, Heidelberg, Germany) tool (E-value < 1 × 10^−5^) [35]. The GO database classifies the functional annotations of these unigenes into three categories: molecular function, biological process, and cellular component. The annotation of the KEGG database matched unigene functions to various biological metabolic pathways, including carbohydrate, lipid, amino acid, and secondary metabolite metabolism pathways. In addition, TBtools was used to assist in the analysis of functional annotations of unigenes [36].

### 2.5. EST-SSR Primer Design

Unigenes carrying SSRs and annotated to metabolic pathways were consequently used as templates to design EST-SSR markers according to the 100 bp sequences flanking the SSR loci using Primer3 (version 2.4.0, San Diego, CA, USA), as previously reported [37]. The software was run with the following parameters: primer length, melting temperature, GC content, and amplification product length ranging from 18 to 23 bp, 50° to 65°, 40% to 60%, and 100 to 300 bp. Only one primer pair was retained for each SSR locus in each unigene.

### 2.6. DNA Extraction and Verification of EST-SSR Markers

Young leaves of 45 accessions of *A. trifoliata* were collected for genomic DNA extraction according to the previously described CTAB protocol [38]. To verify the effectiveness and polymorphism of the primers, 100 pairs of primers for DNRs or TNRs were randomly selected and synthesized (Appendix A). They were then amplified by PCR on a PTC-200 thermocycler (MJ Research, Watertown, MA, USA) with a total reaction volume of 25 μL, containing 1.5 mmol/L MgCl_2_, 0.2 mmol/L dNTPs, 1 unit of Taq DNA polymerase, 200 nmol/L of each primer, and 80 ng of template DNA. The PCR conditions were set as follows: 94 °C for 2 min; 32 cycles of 94 °C for 45 s, 55–60 °C for 30 s, and 72 °C for 30 s; and 72 °C for 10 min. PCR products were subsequently separated and visualized on a 6% nondenaturing polyacrylamide gel.

## 3. Results

### 3.1. De Novo Assembly of Transcriptome Sequencing Data and Identification of SSRs

All RNAs were sequenced by Illumina sequencing technology, and the raw transcriptome data (271.49 Gb of clean data) were deposited in the National Centre for Biotechnology Information (NCBI) with accession numbers SAMN16551931–33, SAMN16551934–36, SAMN16551937–39, and SAMN16551940–42 for the young, enlargement, coloring, and mature stages, respectively, and further de novo assembled. A total of 416,363 (323.45 Mb), 463,756 (350.58 Mb), and 491,680 (377.16 Mb) nonredundant genes (unigenes) were obtained and carried SSR loci from the flesh, seed, and rind tissues, respectively; we found 119,394, 132,582, and 140,247 unigenes in the flesh, seed, and rind tissues, respectively. Further analysis showed that 175,604, 194,370, and 207,906 SSRs with 19,844, 22,367, and 23,888 compound forms were detected in the flesh, seed, and rind tissues, respectively. The SSRs in the expressed sequences of the fruit tissue of *A. trifoliata* were abundant, and the frequency of occurrence was 549.53 per Mb. The assembled unigenes from different fruit tissues had similar GC contents and average lengths (Table 1). In addition, the proportion of unigenes carrying SSRs was also remarkably similar among various tissues (Table 1).

### 3.2. Frequency and Distribution of SSR Motifs in Three Fruit Tissues

Both the motif profile and proportion of SSRs were extremely similar in the three *A. trifoliata* fruit tissues, and the number of SSRs sharply declined as the repeat unit length increased, among which short motifs, including SNRs, DNRs, and TNRs, accounted for more than 98% of the total SSRs, while long motifs, including TtNRs, PNRs, and HNRs, composed a very small proportion (<1.5%) (Figure 1a; Appendix A). Further comparison showed that SNRs had the largest proportion with an average of 60.63%, followed by DNRs with an average of 28.39% and TNRs with an average of 9.69% in the three fruit tissues. In addition, different repeat motifs exhibited various proportions within the same repeat unit length. A/T was the most common repeat type of SNR, accounting for an average of 95.57% of SNRs; AG/CT and AT/AT composed an average of 64.17% and 21.50% of DNRs, respectively; among TNRs, the proportions of AAG/CTT, AAT/ATT, and ATC/ATG motifs were 26.03%, 22.26%, and 15.29%, respectively. Finally, we also found that the difference in both the composition and the proportion of repeat motifs in various tissues at distinct stages was small. For example, in 12 samples (three tissues and four developmental stages), the proportion of A/T in SNRs ranged from 95.18% to 95.95% (Figure 1b); the proportions of AG/CT and AT/AT in DNRs ranged from 61.90% to 67.27% and 19.28% to 23.22%, respectively; and those of AAG/CTT, AAT/ATT, and ATC/ATG in TNRs ranged from 24.33% to 27.28%, 20.65% to 23.86%, and 14.53% to 15.82%, respectively (Appendix A).

### 3.3. Expression Differences in TNRs in Different Fruit Tissues and Different Developmental Stages

TNRs detected in expressed sequences are a class of microsatellites that may directly alter gene coding and are compacted in coding regions due to the nature of translation and the dependence of translation on triplet codons [39]. In this study, an analysis of variance on 60 TNRs (excluding AAA, TTT, CCC, and GGG) expressed in *A. trifoliata* fruit tissues showed that 18 (30%) TNRs were differentially expressed in various tissues and developmental stages (Table 2). Among them, 15 (83.33%) TNRs had differential expression among different tissues, and the motifs putatively encoded nine different amino acids, including phenylalanine (TTC), leucine (CTT; TTA), tryptophan (TGG), valine (GTT), histidine (CAC; CAT), glutamine (CAA), proline (CCT), serine (AGT, TCA, TCC, TCT), and asparagine (AAC, AAT), while the other three (16.67%) TNRs, ACA, GCC, and GTA, putatively encoded threonine (ACA), valine (GTA), and alanine (GCC), respectively, and exhibited differential expression at various developmental stages (Table 2).

Chi-square tests for two corresponding TNRs on complementary strands showed that among 30 potential combinations of the 60 TNRs, only two combinations (ATA/TAT and CCT/GGA) were expressed with the expected 1:1 ratio in all samples. In contrast, 11 combinations significantly deviated from the ratio in all samples, and 8 combinations also significantly deviated from the ratio in some samples at the *p* = 0.05 level. Statistical tests of the remaining 9 combinations were not executed because the expression level was too low (Figure 2; Appendix A). This suggested that the expression preference of complementary motifs varied widely in specific tissues and developmental stages. For instance, the complementary motifs AGA (Arg) and TCT (Ser) had significant expression preferences in both rind and seed tissues at all developmental stages, but the preference in flesh only existed at some developmental stages (Figure 2). The expression preference of the complementary motifs ACA (Thr) and TGT (Cys) existed in all tissues at young stages, but it disappeared at later stages of development (Figure 2).

### 3.4. Functional Annotation of Unigenes Containing SSRs

To clarify the function of the unigenes carrying SSRs in the fruit tissues of *A. trifoliata*, a total of 392,223 unigene sequences consisting of 119,394 from flesh, 132,582 from seeds, and 140,247 from rind were BLAST searched in the GO and KEGG databases. A total of 91,238 unigenes containing SSRs were annotated to 529,711 different terms by Gene Ontology (GO) analysis, and they were grouped into three large categories: cellular component (97,510, 18.41%), molecular function (99,115, 18.71%), and biological process (333,086, 62.88%) with 3, 18, and 28 subcategories, respectively (Figure 3a). Among the three types of cellular component categories, the GO terms involved in the cellular anatomical entity type were the most enriched (80,340, 82.39%), followed by the protein-containing complex in the cellular component category (17,160, 17.60%) (Figure 3a). Likewise, among 18 types of molecular function categories, the top 3 types were catalytic activity (39,000, 39.35%), binding (32,636, 32.93%), and transcription regulator activity (8871, 8.95%), and among 28 types of biological process categories, cellular process (71,572, 21.49%), metabolic process (58,038, 17.42%), and biological regulation (36,799, 11.05%) were the three most abundant types (Figure 3a).

The results of KEGG analysis showed that 22,687 (5.78%) out of 392,223 unigenes containing SSRs were assigned to 41,757 (flesh: 12,581, seed: 14,009, rind: 15,167) annotated metabolic pathways (Appendix A), and they were classified into 12 categories, among which carbohydrate (flesh: 3031, seed: 3392, rind: 3554), amino acid (flesh: 1903, seed: 2092, rind: 2321), and lipid metabolism (flesh: 1697, seed: 1852, rind: 1930) were the three largest categories in terms of number (Figure 3b). In addition, many other valuable metabolic pathways, such as nucleotide, terpenoid, and polyketide metabolism, were found in the fruit tissue of *A. trifoliata* (Appendix A).

### 3.5. Development of Functional Molecular Markers

Among 22,687 unigenes carrying SSRs and annotated into KEGG metabolic pathways, a total of 12,114 (53.40%) unigenes could be used to design the primers for EST-SSRs, while the remaining 10,573 (46.60%) unigenes were not suitable for developing EST-SSR markers due to the short flanking sequences, although they also contained SSRs. A total of 16,869 EST-SSR primer pairs were successfully designed from 12,114 unigenes, in which 3017 (24.91%) unigenes could be used to design more than one EST-SSR primer pair because they carried more than one SSR (Appendix A). This batch of functional EST-SSR markers was predictably associated with 12 classes of metabolic pathways based on KEGG functional annotation of unigenes. Among 16,869 primer pairs involved in 23,102 metabolic pathways, 4993 (21.61%) were related to carbohydrate metabolism, 3193 (13.82%) primer pairs were related to lipid metabolism, 2919 (12.64%) primer pairs were related to amino acid metabolism, 1298 (5.62%) primer pairs were related to biosynthesis of other secondary metabolites, and 1171 (5.07%) primer pairs were related to metabolism of terpenoids and polyketides (Appendix A). In addition, we noted that 16,869 EST-SSR markers mainly consisted of 10,034 (59.48%) SNRs, 3628 (21.51%) DNRs, and 1659 (9.83%) TNRs, while only 153 TtNRS, 4 PNRs, and 3 HNRs were identified; the other 1388 (8.23%) markers were combined SSR types.

### 3.6. Validation of EST-SSR Markers

A total of 100 pairs of EST-SSR primers were randomly selected from the primers derived from DNRs and TNRs (Appendix A), in which 94 primers successfully amplified clear bands and 72 primers produced polymorphic bands in a population consisting of 45 accessions of *A. trifoliata* (Appendix A). A total of 270 alleles were detected using 72 polymorphic primer pairs, and the average number of alleles and polymorphic information content of each locus was 3.75, ranging from 1 to 9, and 0.396, ranging from 0 to 0.831, respectively.

## 4. Discussion

### 4.1. Gene Expression Profile of Various Tissues of A. trifoliata Fruit

*A. trifoliata* is a newly domesticated horticultural crop widely cultivated in Southeast Asia due to its multiple commercial purposes. The great economic value of *A. trifoliata* is primarily embodied in its fruits, and specific fruit tissues have various commercial utilizations. For example, the rind is an important component of medicinal plants because it is rich in secondary metabolites, such as saponins and flavones [2], while the flesh is the main edible part of this delicious fresh fruit due to its rich nutritional content, including vitamins, sugars, minerals, and free amino acids [2]. The seeds are the tissue of high interest for cultivating *A. trifoliata* as an oil crop [4]. Therefore, outlining the gene expression profile in different tissues of *A. trifoliata* fruit is valuable for effectively mining functional genes and improving traits of commercial interest. In the present study, the total RNA of three tissues (rind, flesh, and seed) of *A. trifoliata* fruit at four different developmental stages (young, enlargement, coloring, and maturity) was sequenced by Illumina sequencing technology, and a total of 271.49 Gb of raw transcriptomic data were produced and submitted to NCBI. The available data have been downloaded to study the expression profile of functional genes and repeat sequences by various authors [20,31]. The large amount of expression data provides abundant molecular information for further identifying functional genes involved in specific metabolic pathways and developing EST markers, which could accelerate genetic improvement by molecular marker-assisted selection.

### 4.2. Distribution, Composition, and Expression Profile of SSR Loci in A. trifoliata Fruit Tissues

Functionally, SSRs were initially regarded as “junk” and “neutral” DNA sequences, so the traditional view was that they represented chromosomal regions with few functional genes [40]. However, with the development of whole-genome sequencing technology, genomic data analysis of many species found that there is high SSR density in the flanking regions of functional genes [41,42], which indicated that SSRs could be contained anywhere in chromosomes. In fact, some studies have confirmed that SSRs widely exist in coding and noncoding regions [43,44]. Expressed sequences also contain a large number of SSRs, and moreover, they can exhibit differential expression levels, similar to functional genes.

We found that SSRs were highly enriched in the expressed sequences of various fruit tissues at various developmental stages in *A. trifoliata*. Comparative analysis of the expression of SSRs in *A. trifoliata* fruit tissues revealed that the expression frequencies of different SSRs were similar in different fruit tissues and developmental stages, the proportion of dominant SSRs was very consistent (Figure 1b–d), and the average density of EST-SSRs was 549.53 per Mb (Table 1). A previous study showed that the average density of g-SSRs was 665.28 per Mb in the *A. trifoliata* genome [20], which suggested that the SSR density of coding regions could generally be lower than that of noncoding regions. In addition, in all three fruit tissues, the proportion of unigenes carrying SSRs was more than 28.00%, which was noticeably higher than that in rice (4.7%) [45], maize (1.5%) [45], wheat (7.41%) [46], and radish (7.93%) [47]. Therefore, it is evident that many genes in the *A. trifoliata* genome contain one or more SSR loci.

In addition, the proportion (98.71%) of short repeat unit SSRs (SNRs, DNRs, and TNRs) in the transcriptome was exceptionally close to that (98.67%) in the *A. trifoliata* genome, and the proportion (60.63%) of SNRs in the transcriptome was obviously lower than that (66.98%) in the genomic data. In contrast, the proportion (9.69%) of TNRs in the transcriptome was obviously higher than that (5.60%) in the genome [20], which further supported the view that most SSRs in the coding region are TNRs [22,41]. Previous studies suggested that SSRs in coding regions could be widely subjected to purifying selection, during which only TNRs could be saved due to the nature of translation and triplet codon slippage of the open reading frame [47,48,49]. At the same time, the expression abundance of some SSRs is related to the evolution of organisms [50]. For example, AT/TA usually has a lower expression frequency than other motifs of DNRs in plants, especially in dicots [51]. In this study, we also found that there was some variance in the motif composition of the same repeat unit between transcriptomic and genomic data. For example, AT/TA (53.35%) was the dominant motif of DNRs in the genome [20], while AG/CT was the most abundant (64.17%) motif of DNRs in the expressed sequences (Appendix A). Similarly, in the genome, AAT/ATT accounted for 45.19% of TNRs [20], while it only accounted for an average of 22.26% of TNRs in expressed sequences (Appendix A). Some reports have found that the preference for distinct types of SSRs in functional regions of genes is also wide in plants [42,52]. Therefore, these results indicated that the motif composition of the same repeat unit could be different between genic and intergenic regions.

### 4.3. The Expression of TNRs Is Spatiotemporally Specific in Fruit Tissues of A. trifoliata

We further focused more attention on the differential expression profile of TNRs because they are usually more common than other repeat units in coding regions [53,54]. In the present study, among 60 TNR motifs, 15 motifs exhibited differential expression in specific fruit tissues, while three motifs (ACA, GCC, and GTA) had a differential expression level at selected developmental stages (Table 2), which showed that tissue-specific and developmental stage-specific expression of TNRs is common in *A. trifoliata*, and this differential expression could directly couple with the differential expression of functional genes.

Patterns of DNA double-strand transcription mainly include asymmetric transcription and antisense transcription [55]. A large number of protein-coding genes follow asymmetric transcription rules so that only one coding strand is transcribed. In contrast, the expression of natural antisense transcripts allows complementary strand sequences to be simultaneously transcribed [56], which can regulate gene transcription by forming double-stranded RNA with the mRNA coding [57,58]. In this study, the results of the chi-square test showed that only two complementary motif combinations (ATA/TAT and CCT/GGA) complied with the 1:1 ratio in all 12 transcriptomes of fruit tissue, while many complementary TNR combinations significantly deviated from the 1:1 ratio in all or some samples (Figure 2; Appendix A), possibly resulting from the asymmetric expression of double-stranded DNA, which could commonly occur in various fruit tissues at different developmental stages. The complementary TNR expression profile afforded useful information about natural antisense transcription, which further helped us to understand gene expression regulation.

### 4.4. Development of a Set of Functional EST-SSR Markers and Their Application Prospects

EST-SSR markers are a class of accessible and versatile functional markers because they are directionally derived from expressed sequences associated with functional genes. Although 9394 SSRs were identified and 100 EST-SSR markers were developed according to the transcriptome data of *A. trifoliata*, the transcriptome data were derived from the leaf (which has low economic value) rather than the fruit (which has high economic value) [59]. The markers had less applicable prospects because only the EST-SSRs derived from the transcriptome data of objective organs could be highly associated with corresponding traits of interest [60,61,62]. In this study, the transcriptome data of three fruit tissues were employed to search for SSR loci in expressed sequences, and we further successfully developed 16,869 EST-SSR primer pairs.

In addition, the 12,114 unigenes deriving 16,869 EST-SSR markers were functionally annotated, among which some were involved in metabolic pathways influencing fruit nutrients, such as carbohydrate (3651), amino acid (2200), and vitamin (1274) metabolism, while others were involved in metabolic pathways influencing oil content and components, such as lipids (2217) and unigenes involved in the biosynthesis of secondary metabolites, such as terpenoids or polyketides (876), glycans (798), and xenobiotics (593) (Appendix A). Therefore, when we tried to improve the nutrient characteristics of *A. trifoliata* as a new delicious fresh fruit, 9637 EST-SSR markers (4993, 2919, and 1725 associated with carbohydrates, amino acids, and vitamins, respectively) should be more powerful than the remaining markers (Appendix A). Likewise, 3193 EST-SSR markers associated with lipid metabolism and 1298 EST-SSR markers associated with secondary metabolites could be more valuable than the other markers when exploiting *A. trifoliata* as an oil crop and medicinal plant, respectively.

Finally, PCR amplification results showed that 94 out of 100 primer pairs produced clear products, and 72 primer pairs exhibited polymorphism, which suggested that the EST-SSR markers developed in this study were dependable and effective. Although the PIC value (0.396) was slightly lower than that of the genomic SSR markers developed according to the genome sequence [20], partly due to the more conserved genic regions compared with intergenic regions across the whole genome [63], the high polymorphism of the set of EST-SSR markers should meet almost all the needs of molecular markers in the genetic improvement field. It was evident that the newly developed 16,869 EST-SSR markers could have large application prospects in the future.

## 5. Conclusions

In conclusion, a complete set of transcriptomic data of three *A. trifoliata* fruit tissues at four developmental stages was published in this study. The identification of SSRs in de novo assembled transcriptome data showed that the fruit tissues of *A. trifoliata* possess abundant SSRs, and the expression profile of the SSRs was outlined. The proportion of the same type of SSR motifs was very similar in different fruit tissues and developmental stages, but the expression levels of some TNRs were different in different tissues or developmental stages. We further developed a total of 16,869 EST-SSRs according to the 12,114 assembled unigene sequences, and their validation was confirmed by PCR amplification. Finally, comparative analysis of expression showed that complementary TNRs widely exhibited tissue- and time-specific expression preferences, which could be related to gene expression regulation by natural antisense transcripts. These 16,869 EST-SSR markers, especially the 1659 (9.83%) TNR EST-SSR markers, could be used as powerful molecular tools for molecular selection, functional gene identification, and gene mining in *A. trifoliata* in the future.

## Figures and Tables

**Figure 1 genes-13-01451-f001:**
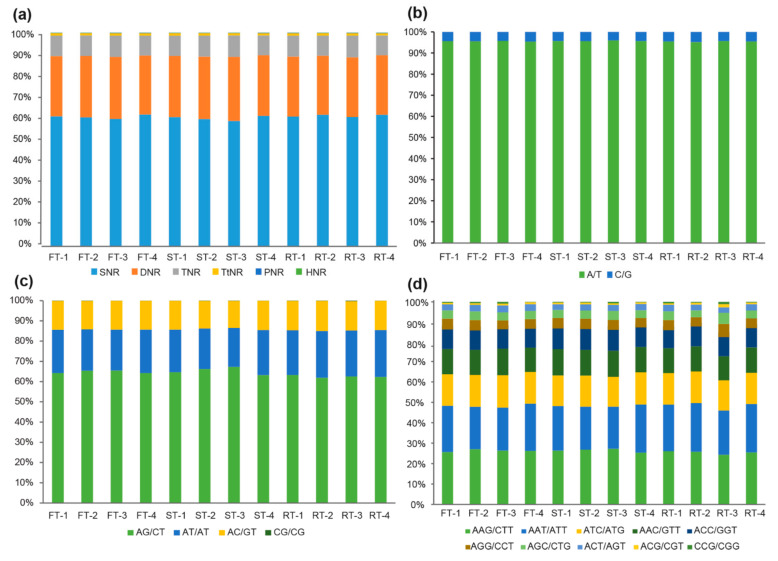
Frequency and distribution of SSR motifs in three fruit tissues of *A. trifoliata* (FT: flesh, ST: seed, RT: rind, -1: young stage, -2: enlargement stage, -3: coloring stage, -4: mature stage). (**a**) The proportion of different repeat unit lengths, (**b**) the proportion of different repeat motifs within SNRs, (**c**) the proportion of different repeat motifs within DNRs, and (**d**) the proportion of different repeat motifs within TNRs.

**Figure 2 genes-13-01451-f002:**
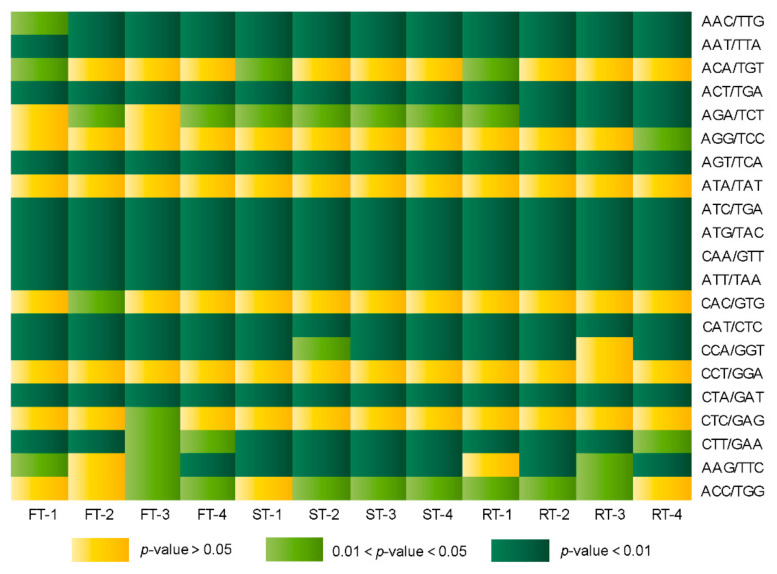
Chi-square test of complementary TNRs.

**Figure 3 genes-13-01451-f003:**
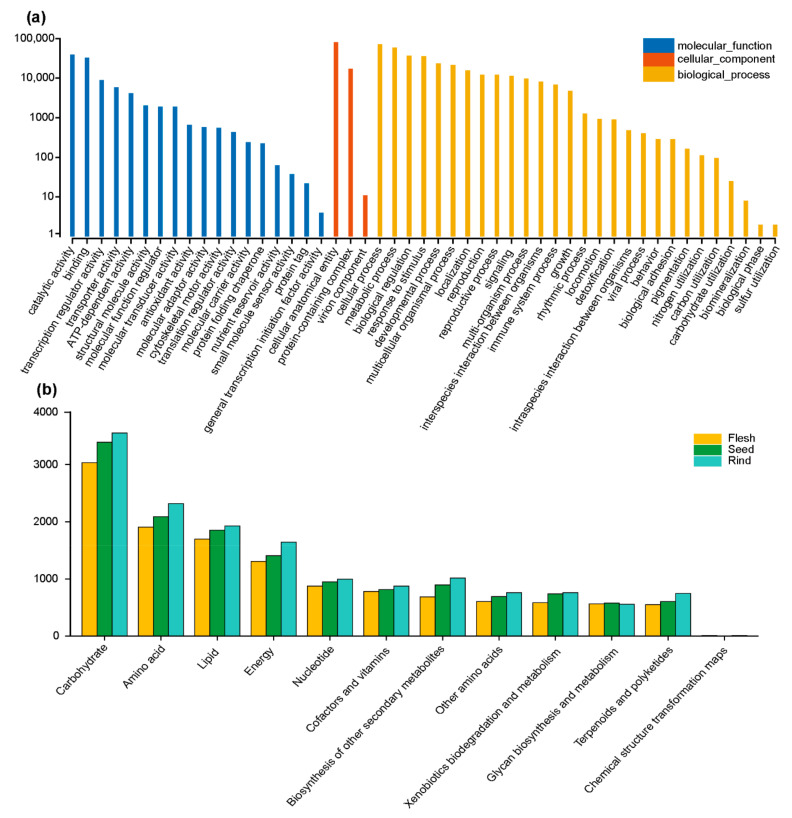
GO and KEGG metabolic pathway annotation of unigenes containing SSRs. (**a**) Distribution of GO classification of unigenes carrying SSRs; (**b**) Distribution of KEGG classification of unigenes carrying SSRs.

**Table 1 genes-13-01451-t001:** De novo assembly results and SSR identification in *A. trifoliata* fruit tissues.

Parameter	Sum (Flesh)	Sum (Seed)	Sum (Rind)
Total number of sequences examined:	416,363	463,756	491,680
Total size of examined sequences (bp):	323,449,301	350,577,642	377,164,182
Total number of identified SSRs:	175,604	194,370	207,906
Number of sequences containing SSRs:	119,394	132,582	140,247
Number of sequences containing more than 1 SSR:	37,747	41,969	45,387
Number of SSRs present in compound formation:	19,844	22,367	23,888
Percent GC (%):	39.37%	39.29%	39.31%
The average length of the sequence (bp):	777.08	756.47	769.8
Mean distribution distance (SSR/Mb):	542.91	554.43	551.24
Proportion of unigenes with SSRs:	28.68%	28.59%	28.52%

**Table 2 genes-13-01451-t002:** Analysis of variance for TNRs in different fruit tissues and developmental stages.

Motif	Amino Acid	Three Fruit Tissues					Developmental Period				
		Sum of Squares	Degree of Freedom	Mean of Squares	F-Value	*p*-Value	Sum of Squares	Degree of Freedom	Mean of Squares	F-Value	*p*-Value
AAC	Asn	554.67	2	277.33	6.84	0.028	124.67	3	41.56	1.03	0.446
AAT	Asn	19,478.17	2	9739.08	6.49	0.032	207.58	3	69.19	0.05	0.986
AGT	Ser	148.17	2	74.08	17.21	0.003	45.67	3	15.22	3.54	0.088
CAA	Gln	1083.17	2	541.58	6.87	0.028	385.67	3	128.56	1.63	0.279
CAC	His	1383.5	2	691.75	7.11	0.026	222.92	3	74.31	0.76	0.554
CAT	His	914.67	2	457.33	6.74	0.029	150.92	3	50.31	0.74	0.565
CCT	Pro	184.67	2	92.33	9.03	0.015	29.67	3	9.89	0.97	0.467
CTT	Leu	578.17	2	289.08	5.55	0.043	158.25	3	52.75	1.01	0.45
GTT	Val	501.17	2	250.58	8.28	0.019	82	3	27.33	0.9	0.493
TCA	Ser	2931.17	2	1465.58	12.52	0.007	39.58	3	13.19	0.11	0.949
TCC	Ser	115.17	2	57.58	9.21	0.015	8.25	3	2.75	0.44	0.733
TCT	Ser	3095.17	2	1547.58	8.4	0.018	885.67	3	295.22	1.6	0.285
TGG	Trp	580.67	2	290.33	10.58	0.011	216.33	3	72.11	2.63	0.145
TTA	Leu	5202	2	2601	8.15	0.019	1187.58	3	395.86	1.24	0.375
TTC	Phe	5552.67	2	2776.33	11.8	0.008	174.92	3	58.31	0.25	0.86
ACA	Thr	132.17	2	66.08	1.85	0.236	689.67	3	229.89	6.45	0.026
GCC	Ala	0.17	2	0.08	0.03	0.969	50.92	3	16.97	6.43	0.026
GTA	Val	2	2	1	0.14	0.872	107.33	3	35.78	5.03	0.045

## Data Availability

All data analyzed during this study are included in the manuscript and Appendix A, and transcriptomic data of *A. trifoliata* fruit tissues have been deposited in the National Center for Biotechnology Information (NCBI) database (https://www.ncbi.nlm.nih.gov/sra (accessed on 29 October 2020)) under accession numbers SAMN16551931–33, SAMN16551934–36, SAMN16551937–39, and SAMN16551940–42.

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
