# Peer review of "Expression Profiles of Microsatellites in Fruit Tissues of Akebia trifoliata and Development of Efficient EST-SSR Markers"

_genes, 2022, doi:10.3390/genes13081451_

Round 1

Reviewer 1 Report

Overall the present manuscript prepared and presented very nicely, the authors presented huge information in support of their findings. There are very few reports available about marker development on Akebia trifoliata but there are numerous grammatical mistakes, and the use of unscientific words throughout the manuscript. The language must be improved.

Author Response

We are grateful to the reviewers for their positive comments on this manuscript. The language of the manuscript has been comprehensively improved under the help a professional company “American Journal Experts”.

Reviewer 2 Report

In the current manuscript, Chen et al., elucidated the transcriptome data of three tissues, including the rind, flesh, and seed, of A. trifoliata fruit at four different developmental stages using Illumina HiSeq technology; they were then employed to isolate and identify candidate unigenes containing SSRs. Both motif and expression profiles were also outlined, and expression differences in TNRs in different tissues at different stages were highlighted. Indeed, the main objective of the study was to develop and validate a set of highly effective EST-SSR markers, which could be valuable for improving traits of interest in A. trifoliata for specific commercial or economic purposes as well as for further exploitation of other wild plants.

The authors identified 175,604, 194,370, and 207,906 SSRs from the de novo assembled 416,363, 463,756, and 491,680 unigene sequences obtained from the flesh, seed, and rind tissues, respectively. The profile and proportion of SSR motifs expressed in each fruit tissue and developmental stage were remarkably similar, but many TNRs had differential expression levels among different tissues or at different developmental stages. In addition, the authors successfully designed 16,869 functional EST-SSR primers according to the annotated unigenes. Finally, 94 and 72 primer pairs of 100 randomly selected primer pairs produced clear bands and polymorphic bands, respectively. These results also elucidated the expression profile of different tissues at various stages. Additionally, the authors provided a set of effective, polymorphic, and reliable EST-SSR markers sufficient to be helpful in accelerating the discovery of metabolic and pathway-specific functional genes for genetic improvement and increased commercial productivity.

Although the topic is attractive, there are some concerns that should be addressed.

-There are some typographical and grammatical errors.

Line 60: Akebia trifoliata is a horticultural crop. Please provide some examples of using molecular markers in horticultural crops such as Paeonia (https://doi.org/10.1007/s00425-019-03299-9), Cannabis (https://doi.org/10.1016/j.indcrop.2020.113026), melon (https://doi.org/10.1016/j.jgeb.2018.08.002), Rosa species (https://doi.org/10.1080/14620316.2021.1894078).

-The quality of Fig.1 is not good. Please replace it with a better one.

- Discussion should be improved.

- The conclusion section is very short. At least it should discuss more future work

Author Response

Point 1: There are some typographical and grammatical errors.

Response 1: We have carefully corrected typographical and grammatical errors in the manuscript, including article substitution, verb tense, preposition usage, etc.

Point 2: Line 60: Akebia trifoliata is a horticultural crop. Please provide some examples of using molecular markers in horticultural crops such as Paeonia (https://doi.org/10.1007/s00425-019-03299-9), Cannabis (https://doi.org/10.1016/j.indcrop.2020.113026), melon (https://doi.org/10.1016/j.jgeb.2018.08.002), Rosa species (https://doi.org/10.1080/14620316.2021.1894078).

Response 2: Thank you very much for your advice. In this section, we simplified the use of molecular markers in cereal crops and emphasized the application of molecular markers in fruit trees. In addition, we had added the corresponding reference. 

Point 3: The quality of Fig.1 is not good. Please replace it with a better one.

Response 3: Fig. 1 had been replaced with a higher quality and clearer image.

Point 4: Discussion should be improved.

Response 4: We have adjusted and revised the discussion to make the logic of each part of the discussion more rigorous. For example, the characterization of SSRs in various tissues of Akebia trifoliata fruit was moved from Section 4.3 to 4.2. In addition, we have further revised some sentences and added references.

Point 5: The conclusion section is very short. At least it should discuss more future work

Response 5: We further expanded the conclusion section, adding conclusions on the composition, distribution and expression characteristics of SSRs in various tissues of Akebia trifoliata, and discussed the future application of the SSR markers developed in this study.

Reviewer 3 Report

Reviews for manuscript (genes-1859310)
This is an important study. The authors have identified simple sequence repeats (SSRs) from transcriptomic data of Akebia trifoliata from 12 samples at different developmental stages. The methods are well designed, results are clearly presented, and the manuscript reads well; however, there are some issues that the authors need to address. Please see my comments:
1. The title is very long and needs to be shortened.
2. In Table 1, I suggest updating the table to compute the frequency (SSR/Mb) and density (SSR/bp) for the total number of identified SSRs for each tissue.
3. Please check the tools used in this study and add the tool version.

Author Response

Point 1: The title is very long and needs to be shortened.

Response 1: We further summarized and shortened the title, which was changed to Expression profiles of microsatellites in fruit tissues of Akebia trifoliata and development of efficient EST-SSR markers

Point 2: In Table 1, I suggest updating the table to compute the frequency (SSR/Mb) and density (SSR/bp) for the total number of identified SSRs for each tissue.

Response 2: Thank you very much for your advice. We updated the frequency of SSR in Table 1 and the manuscript text, but in Table 1 we only kept the frequency due to the similar meaning of frequency and density

Point 3: Please check the tools used in this study and add the tool version

Response 3: We carefully checked the tools in the manuscript and added tool versions.

Round 2

Reviewer 2 Report

All my comments have been addressed. I think that the current form of MS can be published in Genes.